# On the Adversarial Robustness of Neural Networks without Weight Transport

**Mohamed Akrout**
University of Toronto, Triage

## Abstract

Neural networks trained with backpropagation, the standard algorithm of deep learning which uses weight transport, are easily fooled by existing gradient-based adversarial attacks. This class of attacks are based on certain small perturbations of the inputs to make networks misclassify them. We show that less biologically implausible deep neural networks trained with feedback alignment, which do not use weight transport, can be harder to fool, providing actual robustness. Tested on MNIST, deep neural networks trained without weight transport (1) have an adversarial accuracy of 98% compared to 0.03% for neural networks trained with backpropagation and (2) generate non-transferable adversarial examples. However, this gap decreases on CIFAR-10 but is still significant particularly for small perturbations of magnitude less than ½.

## 1   Introduction

Deep neural networks trained with backpropagation (BP) are not robust against certain hardly perceptible perturbation, known as adversarial examples, which are found by slightly altering the network input and nudging it along the gradient of the network's loss function [1]. The feedback-path synaptic weights of these networks use the transpose of the forward-path synaptic weights to run error propagation. This problem is commonly named the *weight transport* problem.

Here we consider more biologically plausible neural networks introduced by Lillicrap et al. [2] to run error propagation using feedback-path weights that are not the transpose of the forward-path ones i.e. without weight transport. This mechanism was called *feedback alignment* (FA). The introduction of a separate feedback path in [2] in the form of random fixed synaptic weights makes the feedback gradients a rough approximation of those computed by backpropagation.

Since gradient-based adversarial attacks are very sensitive to the quality of gradients to perturb the input and fool the neural network, we suspect that the gradients computed without weight transport cannot be accurate enough to design successful gradient-based attacks.

Here we compare the robustness of neural networks trained with either BP or FA on three well-known gradient-based attacks, namely the fast gradient sign method (FGSM) [3], the basic iterative method (BIM) and the momentum iterative fast gradient sign method (MI-FGSM) [4]. To the best of our knowledge, no prior adversarial attacks have been applied for deep neural networks without weight transport.

## 2   The weight-transport problem and its feedback alignment solution

A typical neural network classifier, trained with the backpropagation algorithm, computes in the feedback path the error signals $\boldsymbol{\delta}$ and the weight update $\Delta\mathbf{W}$ according to the error-backpropagation equations:

$$\begin{aligned}
\boldsymbol{\delta}_l &= \phi'(\mathbf{y}_l)\,\mathbf{W}_{l+1}^T\,\boldsymbol{\delta}_{l+1} \\
\Delta\mathbf{W}_l &= \eta_W\,\mathbf{y}_{l-1}\boldsymbol{\delta}_l
\end{aligned} \tag{1}$$

where $\mathbf{y}_l$ is the output signal of layer $l$, $\phi'$ is the derivative of the activation function $\phi$ and $\eta_W$ is a learning-rate factor.
For neuroscientists, the weight update in equation 1 is a biologically implausible computation: the backward error $\boldsymbol{\delta}$ requires $\mathbf{W}^T$, the transposed synapses of the forward path, as the feedback-path synapses. However, the synapses in the forward and feedback paths are physically distinct in the brain and we do not know any biological mechanism to keep the feedback-path synapses equal to the transpose of the forward-path ones [5, 6].

Submitted to the Real Neurons & Hidden Units Workshop @ NeurIPS 2019. Do not distribute.

To solve this modeling difficulty, Lillicrap et al. [2] made the forward and feedback paths physically distinct by fixing the feedback-path synapses to different matrices $\mathbf{B}$ that are randomly fixed (not learned) during the training phase. This solution, called *feedback alignment*, enables deep neural networks to compute the error signals $\boldsymbol{\delta}$ without weight transport problem by the rule

$$\boldsymbol{\delta}_l = \phi'(\mathbf{y}_l) \, \mathbf{B}_{l+1} \, \boldsymbol{\delta}_{l+1} \tag{2}$$

In the rest of this paper, we add the superscript "bp" and "fa" in the notation of any term computed respectively with backpropagation and feedback alignment to avoid any confusion. We call a "BP network" a neural network trained with backpropagation and "FA network" a neural network trained with feedback alignment.

Authors in [7] showed recently that the angles between the gradients $\Delta \mathbf{W}^{fa}$ and $\Delta \mathbf{W}^{bp}$ stay $> 80°$ for most layers of ResNet-18 and ResNet-50 architectures. This means that feedback alignment provides inaccurate gradients $\Delta \mathbf{W}^{fa}$ that are mostly not aligned with the true gradients $\Delta \mathbf{W}^{bp}$.

Since gradient-based adversarial attacks rely on the true gradient to maximize the network loss function, can less accurate gradients computed by feedback alignment provide less-effective adversarial attacks ?

## 3   Gradient-based adversarial attacks

The objective of gradient-based attacks is to find gradient updates to the input with the smallest perturbation possible. We compare the robustness of neural networks trained with either feedback alignment or backpropagation using three techniques mentioned in the recent literature.

### 3.1   Fast Gradient Sign Method (FGSM)

Goodfellow et al. proposed an attack called Fast Gradient Sign Method to generate adversarial examples $x'$ [3] by perturbing the input $x$ with one step gradient update along the direction of the sign of gradient, which can be summarized by

$$x' = x + \epsilon \, sign\big(\nabla_x J(x, y^*)\big) \tag{3}$$

where $\epsilon$ is the magnitude of the perturbation, $J$ is the loss function and $y^*$ is the label of $x$. This perturbation can be computed through transposed forward-path synaptic weights like in backpropagation or through random synaptic weights like in feedback alignment.

### 3.2   Basic Iterative Method (BIM)

While the Fast Gradient Sign method computes a one step gradient update for each input $x$, Kurakin et al. extended it to the Basic Iterative Method [8]. It runs the gradient update for multiple iterations using small step size and clips pixel values to avoid large changes on each pixel in the beginning of each iteration as follows

$$x^0 = x$$
$$x^{t+1} = Clip_X\bigg( x^t + \alpha \, sign\big(\nabla_x J(x^t, y^*)\big)\bigg) \tag{4}$$

where $\alpha$ is the step size and $Clip_X(\cdot)$ denotes the clipping function ensuring that each pixel $x_{i,j}$ is in the interval $[x_{ij}\text{-}\epsilon, x_{ij}+ \epsilon]$. This method is also called the Projected Gradient Descent Method because it "projects" the perturbation onto its feasible set using the clip function.

### 3.3   Momentum Iterative Fast Gradient Sign Method (MI-FGSM)

This method is a natural extension to the Fast Gradient Sign Method by introducing momentum to generate adversarial examples iteratively [4]. At each iteration $t$, the gradient $\mathbf{g}_t$ is computed by the rule

$$x^0 = x, \; \mathbf{g}^0 = \mathbf{0}$$
$$\mathbf{g}^{t+1} = \mu \mathbf{g}^t + \frac{\nabla_x J(x^t, y^*)}{\|\nabla_x J(x^t, y^*)\|} \tag{5}$$
$$x^{t+1} = x^t + \epsilon \, sign\big(\mathbf{g}^{t+1}\big)$$

## 4   Experiments

All the experiments were performed on neural networks with the LeNet architecture [9] with the cross-entropy loss function. We vary the perturbation magnitude $\epsilon$ from 0 to 1 with a step size of 0.1. All adversarial examples were generated using the number of iterations $n = 10$ for BIM and MI-FGSM attacks and $\mu = 0.8$ for the MI-FGSM attack.

## 4.1 Results on MNIST

The results of the accuracy as function of the perturbation magnitude $\epsilon$ on MNIST are given in Figure 1a. We find that when performing the three gradient-based adversarial attacks (FGSM, BIM and MI-FGSM) on a FA neural network, the accuracy does not decrease and stays around 97%. This suggests that MNIST adversarial examples generated by FA gradients cannot fool FA neural networks for $\epsilon \in$ [0,1] unlike BP neural networks whose accuracy drastically decreases to 0% as the perturbation $\epsilon$ increases.

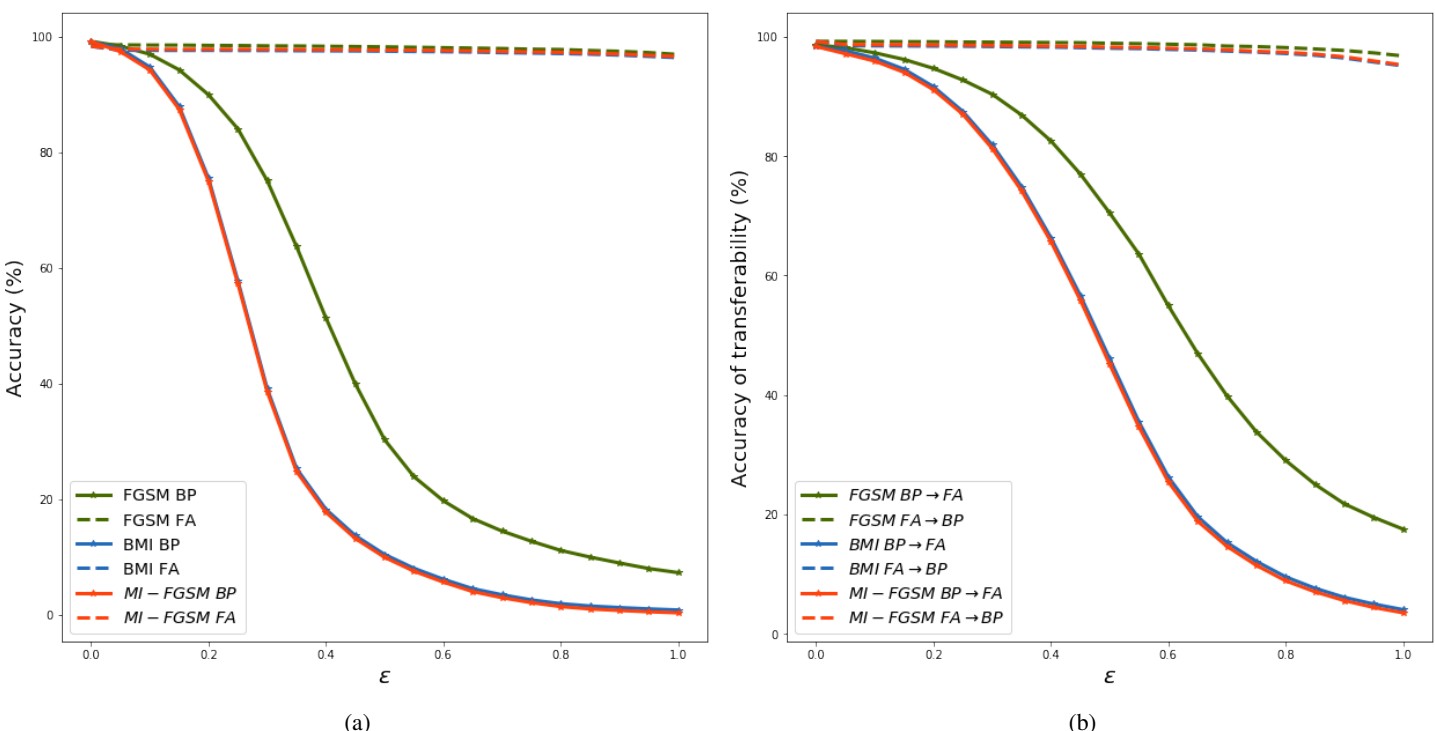

(a)                                                                 (b)

Figure 1: Results on MNIST (a) Adversarial accuracy against three attacks (FGSM, BIM and MI-FGSM) on MNIST for FA and BP networks (b) Adversarial accuracy of transferability between a BP network and a FA network for against three attacks (FGSM, BIM and MI-FGSM) on MNIST. In the legend, we denote by "$BP \rightarrow FA$" the generation of adversarial examples using BP to fool the FA network, and "$FA \rightarrow BP$" the generation of adversarial examples using FA to fool the BP network

Additionally, we investigate the transferability of the adversarial examples generated with either BP or FA networks using each one of the three studied attacks. As shown in Figure 1b, we find that the generated adversarial examples by the FA network don't fool the BP network. This means that these adversarial examples are not transferable. The mutual conclusion is not true since adversarial examples generated by the BP network can fool the FA network.

## 4.2 Results on CIFAR-10

Results on CIFAR-10 about the robustness of FA and BP networks to the three gradient-based adversarial attacks can be found in Figure 2a. Unlike the results on MNIST, the accuracy of FA networks as function of the perturbation magnitude $\epsilon$ does decrease but still with a lower rate than the accuracy of BP networks.

For the transferability of adversarial examples, we still find that the generated adversarial examples by the BP network do fool the FA network. However, unlike the non-transferability of adversarial examples from the FA network to the BP network on MNIST, the BP network is significantly fooled as long as the perturbation magnitude $\epsilon$ increases.

## 5 Discussion and Conclusion

We perform an empirical evaluation investigating both the robustness of deep neural networks without weight transport and the transferability of adversarial examples generated with gradient-based attacks. The results on MNIST clearly show that (1) FA networks are robust to adversarial examples generated with FA and (2) the adversarial examples generated by FA are not transferable to BP networks. On the other hand, we find that these two conclusions are not true on CIFAR-10 even if FA networks showed a significant robustness to

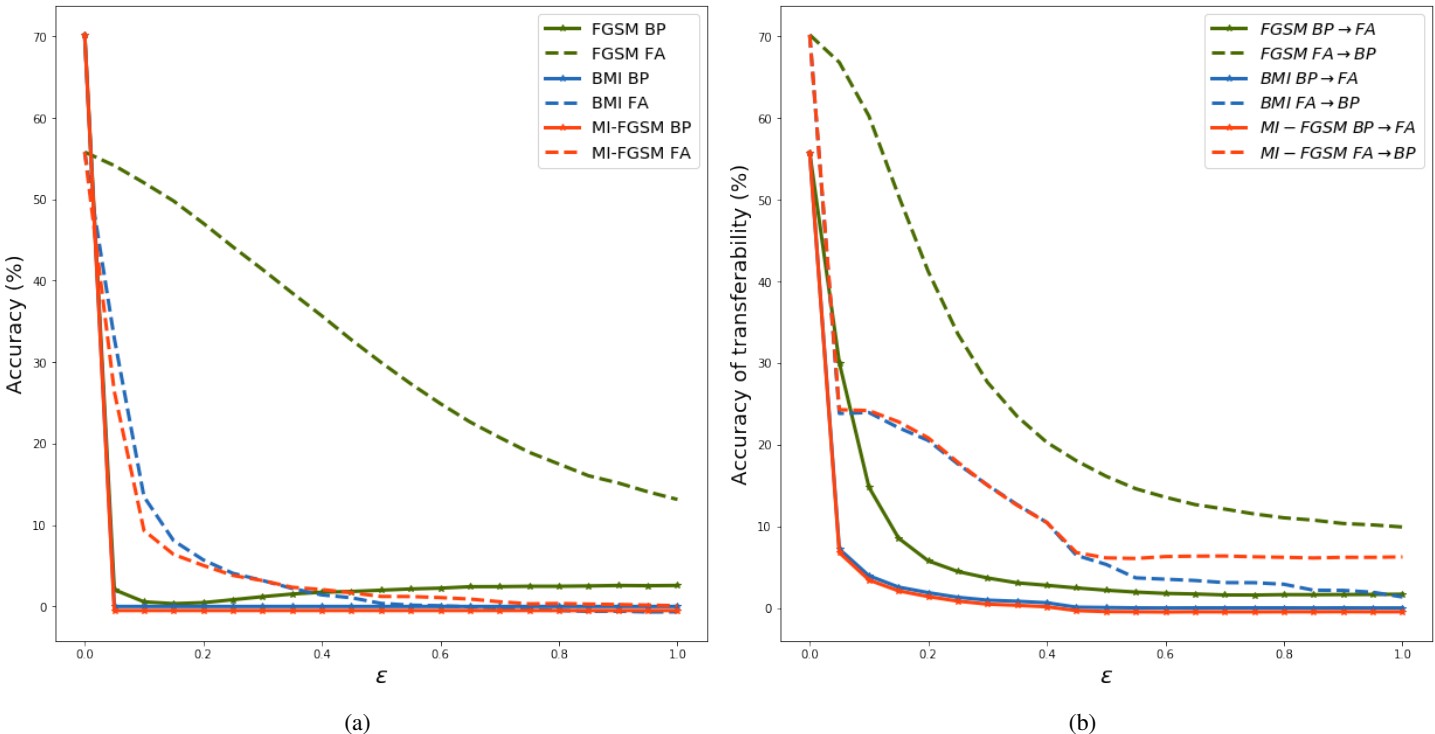

Figure 2: Results on CIFAR-10 (a) Adversarial accuracy against three attacks (FGSM, BIM and MI-FGSM) on CIFAR-10 for FA and BP networks (b) Adversarial accuracy of transferability between a BP network and a FA network on CIFAR-10. Similarly to Figure 1b, we denote by "$BP \rightarrow FA$" the generation of adversarial examples using BP to fool the FA network, and "$FA \rightarrow BP$" the generation of adversarial examples using FA to fool the BP network

gradient-based attacks. Therefore, one should consider performing more exhaustive analysis on more complex datasets to understand the impact of the approximated gradients provided by feedback alignment on the adversarial accuracy of biologically plausible neural networks attacked with gradient-based methods.

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
