# OpenReview forum: "On the Adversarial Robustness of Neural Networks without Weight Transport"
_NeurIPS.cc/2019/Workshop/Neuro_AI — Real Neurons & Hidden Units @ NeurIPS 2019 Poster_

### Official Review · AnonReviewer3 · 2019-09-26
**Sound but premise is strange/unexplained**

**Clarity:** 4

**Comment:**

The premise of the work must be clarified. As well as whether or how adversarial attacks (as framed) might have relevance to neuroscience.

**Category:**

Not applicable

**Clarity Comment:**

No trouble understanding the material or writing

**Evaluation:**

3: Good

**Importance:**

2: Marginally important

**Importance Comment:**

Premise is that feedback alignment networks are also more robust to adversarial attacks. The authors show  because the "gradient" in the feedback pathway is a rough approximation, it is hard to use this gradient to train an adversarial attack.

The basic premise is very strange. Adversarial attacks are artificial: attacker has access to gradient of the loss function. For FA networks, it's unclear why an attacker could not access true gradient, and be forced to use the approximate gradient.

**Intersection:**

3: Medium

**Intersection Comment:**

By focusing on the more biologically plausible "feedback alignment" networks, the paper does sit at the intersection of neuro and AI. However at present, adversarial attacks likely have much larger relevance to AI than neuro.

**Rigor Comment:**

Overall the technical aspects of this paper seem sound.

**Technical Rigor:**

3: Convincing

---

### Official Review · AnonReviewer2 · 2019-09-26
**Interesting idea -- but need more robust testing**

**Clarity:** 4

**Category:**

Neuro->AI

**Clarity Comment:**

The document has been well written.

**Evaluation:**

4: Very good

**Importance:**

3: Important

**Importance Comment:**

This work might open up a new class of neural network model learning framework that could go beyond simply solving adversarial attacks.

**Intersection:**

4: High

**Intersection Comment:**

The work is inspired by a critical difference in feedback connection as applicable to the brain and the models. The author are putting forward a very interesting proposition and it is worth discussing further.

**Rigor Comment:**

It is hard to judge the rigor with such little information. Overall, it seems pretty well managed.

**Technical Rigor:**

3: Convincing

---

### Official Review · AnonReviewer1 · 2019-09-27
**Biologically-plausible learning without weight transport provides robustness to attacks**

**Clarity:** 4

**Comment:**

Overall comments:  Well written paper that explores an interesting idea. The material presented is novel and relevant to the workshop. Experiments conducted do a good job of supporting the authors' claims.

Several small typos:
Line  7   –  “but   is  still”   instead   of   “but   still”  and   “small   perturbations   of   magnitude”   instead   of   “small perturbation magnitude”
Line 34 – Interchange the order of “fa” and “bp”
Line 52 – Kurakin et al
Line 53 – Replace “change” with “changes”
Replace BMI with BIM wherever appropriate.

**Category:**

Neuro->AI

**Clarity Comment:**

Well written.

**Evaluation:**

4: Very good

**Importance:**

4: Very important

**Importance Comment:**

New strategies for learning that use more biologically-plausible learning rules are of extreme importance for the field.

**Intersection:**

4: High

**Intersection Comment:**

neural-inspired learning

**Rigor Comment:**

Results appear to be sound.

**Technical Rigor:**

4: Very convincing

---

### Decision · Program_Chairs · 2019-10-02

Accept (Poster)